# Bond Strength Tests under Pure Shear and Tension between Masonry and Sprayed Mortar

**DOI:** 10.3390/ma13092183

**Published:** 2020-05-09

**Authors:** Dawei Huang, Oriol Pons, Albert Albareda

**Affiliations:** Department of Architectural Technology, Universitat Politècnica de Catalunya, Barcelona 08028, Spain; 13621501383hdw@gmail.com (D.H.); albert.albareda@upc.edu (A.A.)

**Keywords:** shotcrete, structure reinforcement, shear strength, adhesion strength, roughness, JRC-JCS

## Abstract

Sprayed mortar or shotcrete is a construction technology that could enhance existing masonry buildings’ resilience by reinforcing low-safety load-bearing walls. Many factors affect the resistance of shotcrete-reinforced structures. One of the most important is the bond strength at the interface between the shotcrete and the reinforced wall. According to previous technical literature, bond strength usually has two evaluation criteria: shear and tensile strength. The experimental campaign described in this article focused on the bond strength between sprayed mortar and three masonry materials without the influence of normal force or constraint, as well as the roughness of these materials. The analysis of these tests focused on determining the relation between bond strength, roughness, and material strength. The analyses revealed that material strength has a more significant effect on bond strength than roughness, and bond strength is related to shrinkage of the materials. On the basis of previous theories, these researchers found that when there is no obvious influence due to normal force and constraint, the shear strength and tensile strength are different, and the shear strength is likely to be the cohesion force of the two materials. Finally, this article concludes with a novel logarithmic relationship between these strengths.

## 1. Introduction

Many European cities have a high percentage of masonry buildings, most of which can withstand their vertical loads safely but would not be able to withstand any considerable horizontal loads like seismic actions [1]. This is particularly true for cities in low to moderate seismic hazard regions such as Barcelona, where there are thousands of non-heritage masonry buildings from the 1800s and early 1900s [2]. Some of these buildings have very slender masonry load-bearing wall structures, with widths as narrow as 82 mm made of low compressive strength bricks, down to 5 MPa [3]. In these buildings, spraying cementitious materials on the structural masonry walls is a more feasible structural refurbishment solution than using more sophisticated concealed solutions [4].

Sprayed concrete or mortar, known also as shotcrete, is applied using compressed air or another power source to spray the cementitious material through a nozzle onto a surface where the material solidifies and strengthens. Shotcrete was developed in the early 1900s, although there is ongoing research with new findings about its components and application [5,6,7]. Sprayed concrete and sprayed mortar are similar. The main difference between them is the mixture of material. Actually, sprayed concrete evolved from its homonymous mortar [8]. At present, sprayed concrete is mainly used for civil engineering applications such as tunnels [9], while sprayed mortar is basically used for fireproofing steel structures. However, it can also be used to reinforce unreinforced masonry (URM; Table A1 in Appendix A presents a complete list of abbreviations). There have been many studies about existing URM reinforced using sprayed concrete and mortar to obtain earthquake-resistant structures [10,11,12,13].

In order to effectively reinforce masonry walls, sufficient bond strength is required on the brick-concrete or brick-mortar interface. For example, Choi et al. [13] studied the bonding behavior of the joint between sprayed engineered cementitious composite and masonry, and confirmed the contribution of bond strength to wall integrity. However, to the authors’ best knowledge, there is no technical literature on the specific bonding behavior between masonry materials and sprayed mortar. Nevertheless, numerous studies have tested the bond behavior of the interface between joint mortar and brick. Schneemayer et al. [14] studied the fracture-mechanical properties of samples composed of brick and mortar M5 to M30 with different interface surfaces using wedge splitting tests and obtained notch tensile strengths from 0.3 to 2.7 N/mm^2^. Luccioni and Rougier [15] tested the shear strength of the brick–mortar interface in masonry retrofitted with carbon fiber reinforced polymers, following similar procedures to European Standard EN 1052-3 Part 3 to achieve ultimate load values from 5.14 to 23 kN. In addition, many studies that examined the bonding behavior between concrete-concrete, rock-concrete and rock-rock not only measured the bond strength but also quantified the contributions of roughness and adhesion of these bonds. For example, recent research by Saiang et al. [16] and Shen et al. [17], which relied on previous methods developed by Barton & Choubey [18] focusing on rock-rock interfaces, investigated in detail the relation between roughness and shear and tensile strength in the case of rock-concrete joints. Saiang et al. performed direct shear, tensile and compression tests of cylindrical specimens of Ø180 and Ø94, height 140–180 mm composed of fiber reinforced shotcrete and rock, to obtain a joint roughness coefficient (JRC) value from 1 to 13 and a joint compressive strength (JCS) value from 14.0 to 19.0. Shen et al. performed slant shear tests of cubical rock-concrete samples of 100 mm with different interface surface preparations and determined a JRC value from 0.6 to 10.43.

In the field of construction engineering, there are some recent studies on the reliability of discontinuous joint surfaces between concrete or shotcrete and concrete [19,20]. Most studies have indicated that normal strength, friction, and roughness are important factors affecting shear strength. However, most current shear strength tests are carried out by applying a large normal force. According to Barton’s estimation [18], the effect of roughness on shear strength is less obvious as the larger normal stress causes a decrease in stiffness, until only the friction angle has an impact on the shear strength. On the other hand, if there are connectors such as anchors between the shotcrete layer and the element on which the mortar has been sprayed, these connectors constrain the displacement and then the stress increases because of the deformation by the body. These connectors work like a normal force applied on the surface to increase the shear strength. This occurs according to previous studies about the interplay between the dilatancy and shear strength of rock’s discontinuous surface [18]. However, in the absence of connectors or normal forces, which is a case closer to the real environment and to the force characteristics of the sprayed mortar layer, the normal force would be small, close to zero, and there would be no constraint. An important final consequence is that quantification of the shear strength through the envelope obtained from these tests is often not representative. On the other hand, focusing on the mechanical properties of cementitious materials, recent studies have shown that tensile strength is correlated with shear resistance. A study showed that the shear strength is generally greater than tensile strength in the specific case of epoxy polymer concrete [21]. Another study proved that the shear strength is almost equal the tensile strength in the case of low normal stress [16].

In this study, a test was designed and applied without normal force to measure shear stress and tensile stress. Results from different samples were compared, considering roughness and curing time. Subsequently, the shear stress was calculated using existing technology and compared to the test results. This paper is part of a broader research project that studies the application, durability and environmental impact of sprayed concrete and shotcrete for masonry refurbishment. The following sections present previous related research, the experimental campaign, its results, the analysis and discussion of the results and the conclusions.

## 2. Previous Related Research

The bond behavior of mortar joints includes shear and tensile behavior. The Mohr-Coulomb failure criterion is the most effective, accepted constitutive method to describe how bond strength can be effected by normal stress, cohesion and friction angle [22]. Subsequently, Patton [23] pointed out that, except from the effect of the friction angle, the contribution by interface roughness on bond strength is especially significant between rocks due to their usual discontinuous surface. Patton improved the Mohr–Coulomb linear relation by proposing that the average angle of roughness (*i*) and friction angle (*φ*) have a direct effect on the shear strength (*τ*), where *σ* is the normal stress and *c* is the cohesion, as shown below in Equation (1).
(1)τ=σ⋅tan(φ+i)+c

A more recent Barton’s equation for shear strength [18] further clarified the effect of roughness and compressive strength on shear strength by using the JRC/JCS system to define the mathematical relation between normal stress and shear stress. Equation (2) is the basis of this upgrade, where σ is the normal stress, JRC is the joint roughness coefficient and JCS is the joint compressive strength. Both of these coefficients are commonly used parameters in rock analysis.
(2)$$i=JRC\cdot \mathrm{lg}\frac{JCS}{\sigma }$$

Equation (2) can also be used in the case of two different materials, according to an article by Zhang et al. [24] that defined a JCS equation for two materials. To sum up, two types of equations are optimized in this study that apply separately to low and high normal stresses. Equation type II better fits the case study, which is presented below in Equations (3) and (4). In Equation (3), *JCS_a_* is the compression strength of the higher stiffness material and *JCS_b_* is the compression strength of the lower stiffness material.
(3)$$JCS=\left(1\;-\;K_{a}\right)JCS_{a}+K_{a}JCS_{b}$$
(4)$$K_{a}=\frac{\sigma_{ca}}{\sigma_{ca}+\sigma_{cb}}$$

The result of Equation (3) is more accurate the larger the normal stress, according to research carried out by Du et al. [25]. Shen’s group [17] found that the physical adhesion of concrete can be understood as apparent cohesion in the calculation of τ when the Mohr–Coulomb linear criterion is used, because it entails initial and intrinsic shear strength. The adhesion can be assumed as the sum of physical forces between the molecules and chemical attraction [17]. Many factors affect it, such as hydrophilicity and chemical properties, which are also mentioned in other studies [26]. The adhesion strength value can be obtained as the tensile behavior from pull-out tests [16]. By applying a tensile force at both ends of the specimen, the interface tends to break, thereby obtaining the tensile stress for the cross section. This parameter can be difficult to determine, especially when one of the two components fails before the joint collapses.

In contrast, the usual shear stress test method applies a normal force before the horizontal force is applied. The outcome of this test is usually a curve between the displacement of the specimen and the shear stress. The regular procedure for this test is to obtain curves under different normal stresses. Afterwards, a linear regression is usually performed through the peaks of these curves, so that a relationship between normal stress and shear stress can be obtained. This is the method that was used in studies by Saiang et al. [16] and Du et al. [25]. In the present research project, other test methods were sought to improve the aforementioned shear strength test [19,20,23]. To calculate the JRC value in this study, the starting point was the method defined by Du et al. [25]. The following sections introduce some new experiences with JRC calculations.

The bond between sprayed mortar and masonry is not comprised of a single material because it involves at least bricks, joint mortar and sprayed mortar. Therefore, it is not possible to follow Barton’s method directly. As previously explained, research by Zhang et al. [24] introduced a variation to Barton’s method to determine JCS in joints involving different materials. However, JRC could be obtained by compiling a detailed three-dimensional mapping of the roughness of the joint surface. For example, research by Fardin et al. [27] and Alshibli et al. [28] followed these steps to successfully obtain the JRC. To measure the roughness of the joint surface, the most common methods are 3D scanning and mechanical measurement. The 3D scanning records the position information of the rough surface using a 3D scanner. After processing this information, a 3D model is generated, so that all the roughness data is provided in this model. Otherwise, to measure the roughness ratio using mechanical measurements, the profile of the rough surface could be recorded for example using a profilograph and roughness ruler. Then, the JRC can be obtained by compiling these data. Details about the latter methods are described in the article by Du et al. [29].

## 3. Sample Preparation and Experimental Program

### 3.1. Materials and Sample Preparation

In this research project, three typical masonry specimens were used as the base material that was sprayed on. Four materials were used: three bricks, MP1, MP2, and MP3, and one typical mortar for joints, MP4. The three bricks can be currently found on the market with an external waterproof transparent covering layer. The finishing layer obviously affects the MP1 results in the 3 samples and is not present in all masonry elements that require structural refurbishment, but it was the same in all bricks and all ages and, therefore, it did not discriminate the results. MP1 samples are ordinary clay brick with rough surfaces, comprised of relatively uniform clay particles mixed with gravel impurities. MP2 is a high-temperature clay brick, through high-temperature clay fired melting evenly and without impurities. MP3 is a handmade clay brick, rich in impurities, and uneven clay particles. MP4 is mortar for masonry joints with a mixture of 1:3:1, cement R42.5 and sand of 1 mm diameter grains. Table 1 presents the main mechanical properties for these masonry materials. These properties were determined by the manufacturer from tests following EN 771-1:2011 + A1: 2015 [30].

Table 2 depicts the sprayed mortar mix and additives used in this experimental campaign.

Table 3 shows the mortar’s mechanical properties, including compressive strength, density, porosity and absorption. These values were tested on cylindrical samples with a diameter of 25 mm and a height of 50 mm at the age of seven and 28 days.

The samples were constructed in six steps. First, 80 small pieces of each masonry material were prepared by cutting bricks and a mortar panel. These small pieces were 70 × 50 × 15 mm^3^ for MP1 and MP2 and 70 × 30 × 30 mm^3^ for MP3 and MP4. The aforementioned roughness scanning tests were directly performed on these 80 pieces as will be explained in detail in the following section. Second, these pieces were placed evenly in two wooden troughs of 500 × 500 × 150 mm^3^. In order to reduce the effect of shrinkage of the entire body and avoid testing on the defective zone due to the spraying process [31] on the test results, no specimen was placed within the range of 10 cm to the surrounding edge, and the side face of the trough was made at 45 degrees against the base. Third, expanded polystyrene (EPS) pieces were inserted to fill in the spaces between the masonry material pieces where the sprayed mortar could not enter. EPS worked like a mold to ensure that only one surface of the specimen would be in contact with the sprayed mortar. EPS was chosen as the filling material because it was easy to insert and remove, as it had to be taken out after spraying. Figure 1a shows the arrangement of the specimens in the two troughs. Fourth, the wooden troughs were placed at a 30-degree angle on the ground as presented in Figure 1b. Fifth, mortar was sprayed in each trough to obtain the specimens. This layer of sprayed concrete had a thickness of 70–100 mm. The spraying process followed the experimental method developed by Pícolo [32]. Sixth and finally, after a seven-day strengthening period, the two specimens were extracted from the troughs and the EPS was taken out. The previously mentioned pull-out tests were carried out on these two specimens, as explained in depth in the following section.

### 3.2. Experimental Tests

The research campaign had two main parts: roughness and pull-out tests. In the first part, a surface profile gage was used [33], while in the second a manual jack was employed. Both parts relied on high resolution pictures to extract more data for the analysis. This section describes in detail the methods used to carry out these tests.

#### 3.2.1. Roughness Tests

These tests were carried out to obtain the JRC and the arithmetical mean deviation of roughness (R_a_), both of which were used to define roughness. Following the methods of previous studies [29], a horizontal platform was used to allow samples to slide at every 5 mm or 2 mm and record the elevation measurements using a surface profile gage (SPG TS3 Surface Profile Gage, PCE Instruments, DeFelsko, Ogdensburg, NY, USA) (Figure 2a). Figure 2b summarizes this measurement process.

To obtain the parameter for defining *JRC* called *R_A_*, with value Δ*h*/*L*_0_, as shown in Figure 3a and Equation (5), the roughness parameters *L_n_* and *L*_0_ could be measured in the obtained roughness curves. To obtain more accurate results, from each surface three straight roughness lines with measurements every 5 mm in one line and 2 mm in two lines were taken. Finally, JRC was calculated following Equation (5) according to previous research projects [23]:(5)$$\{\begin{array}{c} JRC=49.2114e^{\frac{29L_{0}}{450L_{n}}}\arctan\left(8R_{A}\right)\\ R_{A}=\frac{\Delta h}{L_{n}} \end{array}$$

The calculation of *R_a_* followed Equation (6), which relies on related standards [26] and follows the aforementioned method to measure relevant parameters such as *z*_(*x*)_, *d_x_* and *L* from the roughness curves, as shown in Figure 3b.
(6)$$R_{a}=\frac{1}{L}\overset{L}{\underset{0}{\int }}/\;z_{\left(x\right)}\;/\;d_{x}=\frac{1}{n}\;\overset{n}{\underset{i=1}{\sum }}/\;z_{i}$$

#### 3.2.2. Pull-out Tests

A manual jack (Anchor Tester 28, Hilti, Scaan, Liechtenstein) (Figure 4a) was used in the pull-out tests on the aforementioned two 500 × 500 × 220 mm^3^ specimens obtained from the troughs. These tests were of two types, direct shear and tensile. Before these pull-out tests were performed on each masonry piece, the two specimens had to be located as shown in Figure 4. First, each masonry piece was glued on a bolt with an anchor resin and then this bolt was screwed to the jack to ensure that the connection was secure and vertical. Then, the jack was slowly activated to ensure that the tension was slowly applied to the specimen until the specimen was disengaged. At this moment, the meter was observed to record the peak.

## 4. Experimental Results

This section summarizes the main results of this research project, which are associated with previous explanations about roughness and pull-out tests. The specific results of roughness tests are detailed in Table A2, Table A3 and Table A4 in Appendix B. A comparison of Table A2 results between MP1, MP2 and MP3 revealed that the roughness of MP2 was the lowest, and the JRC and R_a_ values of MP1 and MP3 were very close, although MP3 was rougher and had the highest maximum height of profile (R_y_). Table A2 shows the average curves values for all those measured.

The pull-out tension and shear tests on the 80 masonry material pieces of types MP1 to MP4 sprayed with mortar gave valid results in 76 cases. As control variables, the following two parameters were selected: (1) roughness, with its JRC and R_a_, and (2) specimen age when the tensile tests were carried out, which was either 7 days or 28 days. Table A3 and Table A4 in Appendix B present the data for the 76 specimens and the test results in depth. Table 4 shows a summary of these results.

As presented in Table A3 and Table A4, there were three typical failure modes: (a) disconnection of the bond, (b) fracture of the specimen and (c) fracture by the bond material. The disconnection of the bond occurred when, under the tension of the jack, the specimen and the sprayed mortar were perfectly separated at the interface, and there was no obvious fracture on the surface. This also means that the bond strength was significantly weaker than the resistance of each part of the specimen and sprayed mortar. Moreover, in this case, the values of the test that were measured were very small, and even fell off on their own. Figure 5a shows an example of the smooth surface of the sprayed mortar after the pull-out test with this first failure mode. Figure 5b is a 100 times magnification and shows that, in this failure mode, there are some small cracks on the surface.

The second mode of failure of the specimen that broke in its own section was fracture of the specimen under the tension of the jack. This occurred when it was being separated, as shown in Figure 5c. This failure mode only occurred on MP4 samples. This occurs because the mortar specimen had formed a fragile superficial layer on the surface during the strengthening of the mortar and this outer layer could not withstand the tension. The third mode of failure was fracture of the bond material itself. This mode occurred in both shear and tension pull-out tests. The joint broke due to excessive stress. Therefore, a lot of particles remained on the interface. In shear strength testing, there were significantly more particles remaining on the interface than in tensile/tension testing, and the residual particles were significantly larger or even visible. When the bond strength was relatively large, some parts of the brick were not strong enough, because the material was not uniform. Especially when shear testing, the section of micro interlock had more stress than before. Figure 5e shows, in this third mode of failure, the surface of sprayed mortar where particles remained in it. Figure 5f shows a 100 times magnification where those particles clearly remained. Figure 5c presents some small cracks as in 5a.

## 5. Analysis and Discussion

As stated above, two types of behavior usually need to be considered to analyze the bond strength between mortar and masonry: shear and tensile bond strengths. The aim of the experimental campaign was to provide some values for both bond strengths depending on different masonry-based materials. These values will be useful to designers for general analyses involving sprayed walls. As mentioned before, the response by the bond is crucial to the overall response of these structures. For this purpose, the aforementioned material-based specimens were tested at different stages to detect which significant parameters are involved.

### 5.1. Influence of Compressive Strength

As detailed in Table 1, Table 2 and Table 3, three masonry-based pieces were used in the campaign, all combined with 25.1 MPa sprayed mortar. The bricks showed different normalized compressive strengths of 15, 30 and 45 MPa. As explained in Section 3.2, different tests were carried out under tension and shear using all the aforementioned materials, to determine the influence of compressive strength on the bond response between these bricks and mortar.

Results show how the normalized compressive strength of masonry (f_b_) has a direct influence on the bond response, that is, the tensile bond strength, especially when it is much higher than the mortar strength. This is shown in Figure 6.

For compressive strengths between 15 and 35 MPa, the response was quite similar for tensile and shear strengths, while in the case of 40 MPa the response was completely different. This phenomenon is due to the brittle fracture of the interface of mortar in contact with a tougher material. Figure 6 shows that tensile and shear bond strengths decrease hand in hand with the masonry compressive strength, due to the brittle response of masonry as strength increases. Nevertheless, it seems from the results that the normalized compressive strength of masonry is not significant to the tensile bond behavior if the strengths of both components do not vary widely. It is important to point out that in the cases of 15 and 35 MPa masonry specimens, MP3 and MP1 respectively, results show a wide dispersion between 0.2 and almost 1 MPa. On the other hand, in the case of 40 MPa specimen MP2, the obtained results are much more concentrated in the range between 0 and 0.1 MPa. This fact denotes the brittle fracture of specimens in the case of 40 MPa. Figure 6 also depicts that the shear bond strength is almost double the tensile bond strength. This is due to the roughness between surfaces – its mechanical interlock-, which leads to a brittle fracture for low-bearing masonry that ends in more shear capacity rather than the tensile one. This is related to the parameter JRC as shown in the following sections.

As explained in Section 2, JCS is a specific parameter to define the effect of compressive strength of components on the bond behavior. Comparing the JCS parameters obtained in this experimental campaign to the three brick masonry normalized compressive strengths (f_b_), it can be seen that this parameter is very similar for specimens MP2 and MP3, while it is significantly lower for specimen MP1, as shown in Figure 7. This indicates that the compressive strength of masonry plays an important role in JCS if there is a significant difference between the compressive strengths of both materials, bricks, and mortar, although this is not the decisive variable. A high compressive strength of masonry implies a brittle fracture of sprayed mortar on the interface, resulting in a lower JCS. This parameter remains almost constant up to 35 MPa masonry base, suffering from a sudden descent from that point on. This phenomenon derives from the fact that compressive strength for specimen MP2 is much higher than the strength of mortar (between 21 and 25 MPa), while in cases of MP1 and MP3, both strengths are much more similar to mortar’s capacity. Compatibility of stiffness and strengths for both cited materials are decisive in the tensile or shear bond responses.

### 5.2. Influence of Mortar Age

Specimens were tested at 7 and 28 days under shear and tension to describe the influence of the drying process of mortar on the set. The results presented in Figure 8 and Figure 9 show that tensile adhesion is significantly higher at early ages of mortar rather than after 28 days. This loss of bonding during the aging process is likely to be caused by shrinkage in the sprayed mortar, causing partial bonding behavior failure, which results in decreased adhesion. The loss of strength even reaches 50% in all masonry specimens. At the same time, the second figure shows a clear reduction of shear bond stress beyond 35 MPa masonry specimen.

Once again, the dispersion of results decreases as the compressive strength grows. For 15 MPa masonry specimens of MP3, it is difficult to fix a value for shear bond strength (from 0 to 1.8 MPa), while the range is much more reduced for the cases of 35 and 40 MPa specimens (0.2–0.8 MPa of MP1 samples and 0–0.1 MPa of MP2 specimens respectively).

### 5.3. Influence of Roughness

In contrast to compressive strength, roughness has been identified as a crucial parameter in the definition of bond strength behavior. Results show how roughness has a direct influence on tensile strength, and even more on the shear strength of the interface, as presented in Figure 10. Roughness was analyzed through the well-known parameter JRC, explained in detail in Section 2. By considering the JRC parameter obtained by determining R_a_ in this experimental campaign, two curves for tensile bond strength and cohesion (shear strength) can be drawn. Note there is a loss of about 40% of tensile strength during the mortar drying process.

According to Figure 9, two practical expressions are derived from results for limiting tensile bond strength, depending on the parameter JRC of the interface. The relationship between bond strength and surface roughness (represented by JRC) is different for 7-day than 28-day sprayed mortar. Thus, the two linear Equations (7) and (8) are derived as follows:(7)$$a_{lim}{}_{\left(JRC\right)}=0.418\cdot JRC-0.888,\;\;\mathrm{for}\;7-\mathrm{day}\;25\;\mathrm{MPa}\;\mathrm{mortar}\;\left(R2=0.9830\right)$$
(8)$$a_{lim}{}_{\left(JRC\right)}=0.264\cdot JRC-0.578,\;\;\mathrm{for}\;28-\mathrm{day}\;25\;\mathrm{MPa}\;\mathrm{mortar}\;\left(R2=0.9883\right)$$

A similar approach was used for shear strength. In Shen et al. [17], an attempt was made to combine Barton’s method [18] with the Mohr-Coulomb failure criterion to fit Shen’s test results. Barton’s shear test was always performed under normal pressure. In contrast, in this research, normal strength was almost negligible during the shear test as only the self-weight of the pieces could be considered, as in most real application cases. Thus, according to the Mohr–Coulomb failure criterion, the shear test provided the pure value for cohesion. The parameter of cohesion in the Mohr–Coulomb criteria may be assumed as the pure tangential adherence that depends, at the same time, on roughness. This research defines a linear expression, Equation (9), to define cohesion depending on roughness based on the aforementioned technical literature. See Section 2 and this campaign’s experimental results. In the beginning of this Section 5.3, it has been proved that roughness is the main factor governing bond strength. This Equation (9) aims to refine Equation (1) for the specific case of sprayed masonry walls, by defining an initial value of shear strength depending on the JRC factor in the original Equation (7). This shear strength, in fact, is known as cohesion, as the shear capacity without normal stress is directly the parameter of cohesion. Equation (10) defines factor “*i*” from Equation (9).
(9)$$\tau =\sigma tan(\phi +i)+c_{\left(JRC\right)}$$
where factor “*i*”:(10)$$i=JRC\cdot log_{10}\left(\frac{KJCS_{a}+\left(1-K\right)JCS_{b}}{\sigma }\right)$$

In this new expression, the shear strength capacity of the bond between masonry and sprayed mortar was defined depending on cohesion only, without considering compressive strength or JCS, because it was proved in Section 5.1 that compressive strength of masonry is not crucial in the shear response of the bond up to 35 MPa specimens, by using 29 MPa mortar. Under the conditions in which this campaign was carried out, variable τ is equal to the value of pure cohesion, as has been mentioned before. Therefore, the function which describes cohesion depending on JRC–this is $c_{\left(JRC\right)}$ or shear bond strength-is shown in Figure 11, for 7-day and 28-day cases and derived from the results.

From the aforementioned results, two expressions, namely Equations (11) and (12), for cohesion depending on the JRC factor, or initial shear bond strength in Equation (9), are defined:(11)$$c_{\left(JRC\right)}=0.417\cdot JRC-0.888,\;\;\mathrm{for}\;7-\mathrm{day}\;\mathrm{and}\;25\;\mathrm{MPa}\;\mathrm{mortar}\;\left(R2=0.8736\right)$$
(12)$$c_{\left(JRC\right)}=0.264\cdot JRC-0.578,\;\;\mathrm{for}\;28-\mathrm{day}\;\mathrm{and}\;25\;\mathrm{MPa}\;\mathrm{mortar}\;\left(R2=0.9768\right)$$

This expression complements Equation (1), which had already considered the effect of roughness and compressive strength under normal pressure σ but did not do the same for the initial shear adherence (cohesion), since it is not suitable for rocks. According to the results, the proposed expression is only suitable for masonry materials with bricks fb between 15 and 35 MPa, and only for 25 MPa mortar. For other combinations and assumptions, a broader experimental campaign needs to be carried out.

## 6. Conclusions

The bond strength between masonry walls and reinforcing sprayed mortar is crucial to consider the composite behavior of reinforced URM for proper structural performance. This is a solution which is increasingly being applied to existing buildings to improve the load bearing and stability behavior of various kinds of masonry walls. However, there is no significant literature or experimental campaigns about the bond behavior between the two materials.

Here, a new application of the Barton method is proposed, based on the Mohr–Coulomb failure criterion, through calibrating the results with those of experimental tests under pure shear and tension. Since the analysis is designed for vertical sprayed masonry walls, no normal stresses were considered in the analytical assumptions. Thus, shear stress capacity depends basically on cohesion, which is understood as the adhesion between components, which at the same time depends on roughness and friction.

According to observations, tensile bond strength and cohesion between most samples of brick joint mortar and sprayed mortar are higher than the strength of brick joint mortar. This caused most of the samples in the experiment to break first on the brick joint mortar but not on the interface between them. This is likely to be directly related to the low strength of the brick joint mortar surface. Nevertheless, this should be further verified by future experiments.

The analysis of the results has found that compressive strength of masonry has a direct influence on the tensile bond strength, especially when it is much higher than the mortar strength, but the compressive strength of the bricks is not crucial in the shear response of the bond between 15 and 35 MPa. Research also showed that the tensile bond strength and cohesion of most samples decreases with age, which is likely to be directly related to shrinkage of the material. Further testing is needed to prove this conclusion as well. Results also show how roughness has a direct influence on tensile strength and even more on the shear strength of the interface.

The expressions given in this research on adhesion (tensile bond strength) and cohesion (shear bond strength) are derived directly from the experimental campaign. They are thought to be useful for designers to analyze sprayed masonry walls, as a refinement and adaptation of the Shen et al. method for masonry elements. Former expressions specialized for rocks had already considered the effect of roughness and compressive strength under normal pressure but not for the cohesion since it was not suitable. It is obvious that the proposed expressions are limited to the cases that were studied, involving masonry elements from 15 to 35 MPa and mortar of 25 MPa of compressive strengths. To generalize the expressions, a wider experimental campaign will be needed.

A significant good fit is observed between experimental and analytical results for the shear strength of the bond. Future studies will examine this bond depending on the thickness of each sprayed mortar layer section and the single face or double face configuration and using bricks without waterproof transparent coating.

## Figures and Tables

**Figure 1 materials-13-02183-f001:**
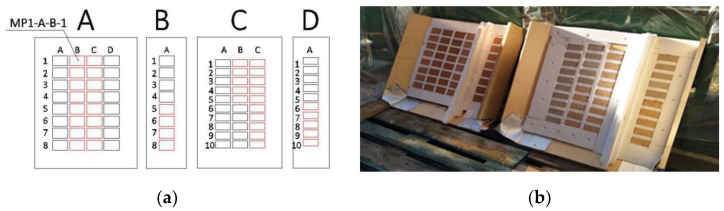
(**a**) Arrangement of the specimens in the two troughs; (**b**) Wooden troughs placed at a 30-degree angle on the ground to be sprayed with mortar.

**Figure 2 materials-13-02183-f002:**
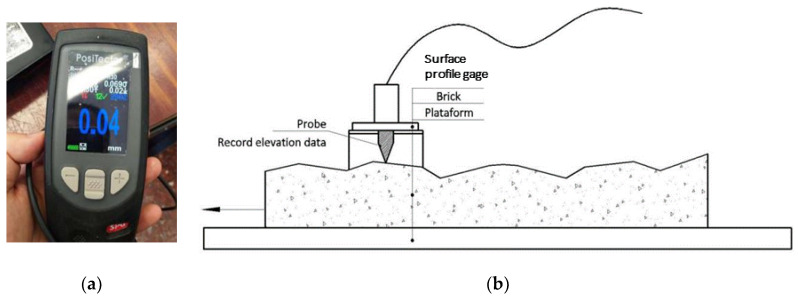
Roughness tests: (**a**) surface profile gage; (**b**) measurement process followed to define surfaces roughness.

**Figure 3 materials-13-02183-f003:**
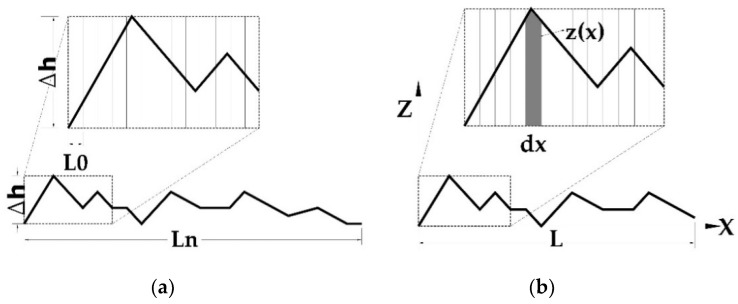
Roughness definition method to define roughness curves: (**a**) roughness parameters L_n_ and L_0_ measured in the roughness curves and (**b**) z_x_ d_x_ and L from the roughness curves.

**Figure 4 materials-13-02183-f004:**
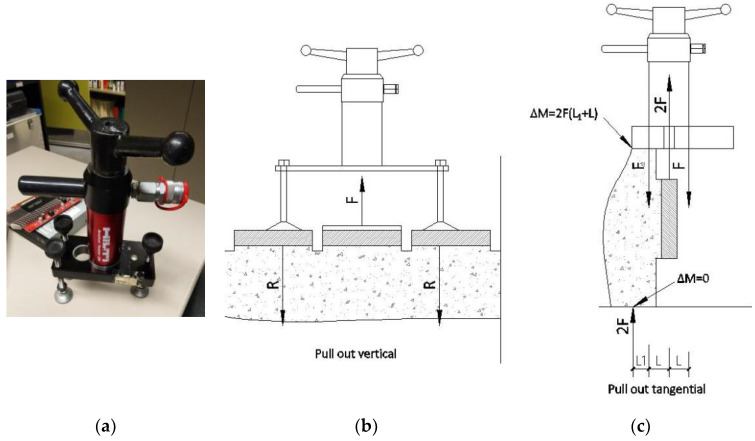
Location of the specimens to carry out the pull-out tests on each masonry piece. (**a**) Manual jack, (**b**) tensile and (**c**) direct shear types.

**Figure 5 materials-13-02183-f005:**
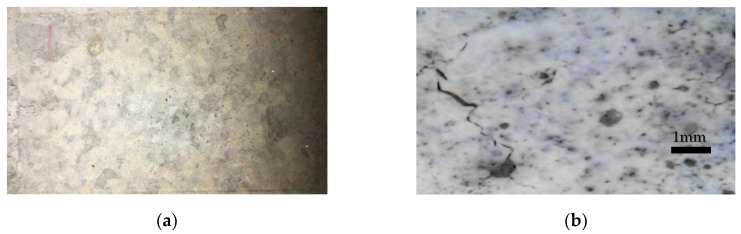
Typical failure modes in pull-out tests: (**a**) disconnection of the bond, (**b**) photo of case (**a**) magnified, (**c**) the specimen broke in its own section (**d**) photo of case (**c**) magnified, (**e**) fracture of the bond material itself, (**f**) photo of case (**e**) magnified.

**Figure 6 materials-13-02183-f006:**
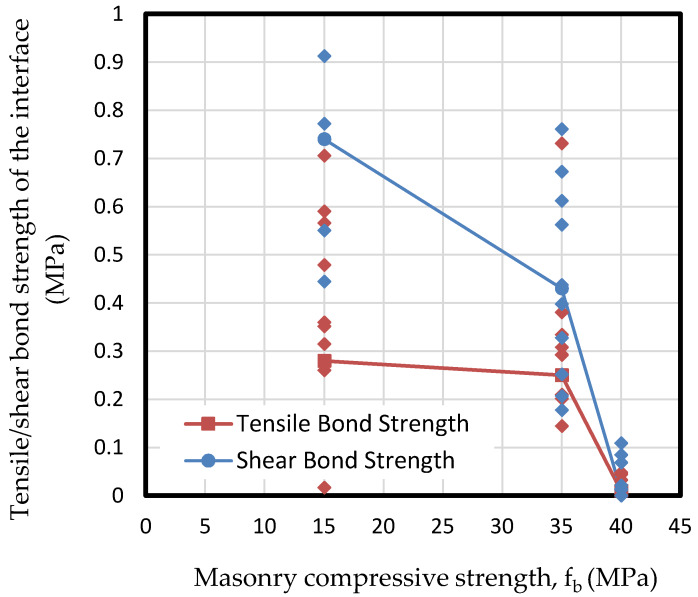
Results of tensile and shear bond strength of the interface depending on the masonry compressive strength (f_b_).

**Figure 7 materials-13-02183-f007:**
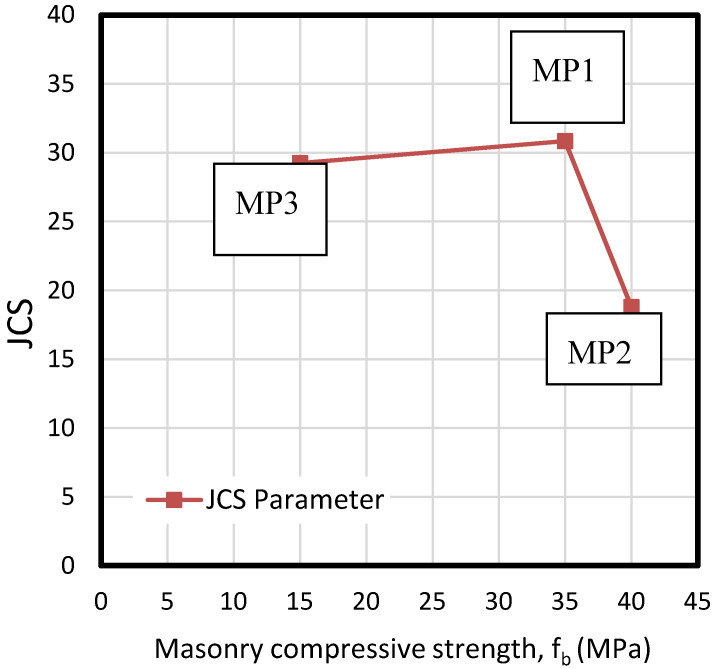
Calculated values of JCS depending on the compressive strength of tested masonry specimens (fb).

**Figure 8 materials-13-02183-f008:**
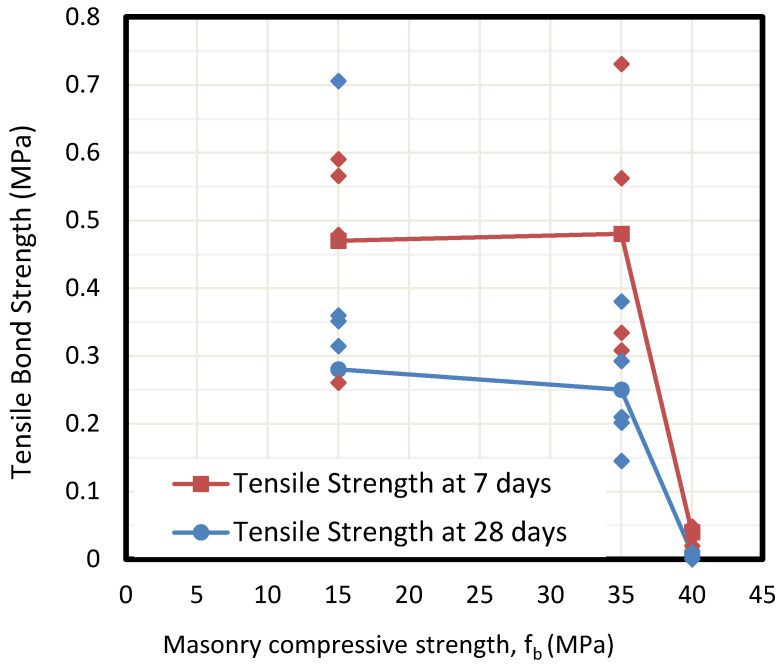
Calculated values of tensile bond strengths depending on bricks (fb) at different ages of mortar drying.

**Figure 9 materials-13-02183-f009:**
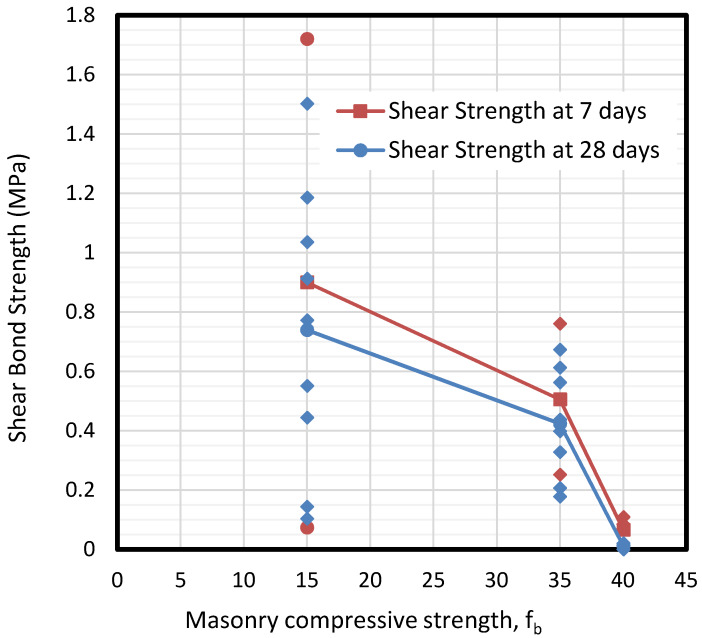
Calculated values of shear bond strengths depending on bricks (fb) at different ages of mortar drying.

**Figure 10 materials-13-02183-f010:**
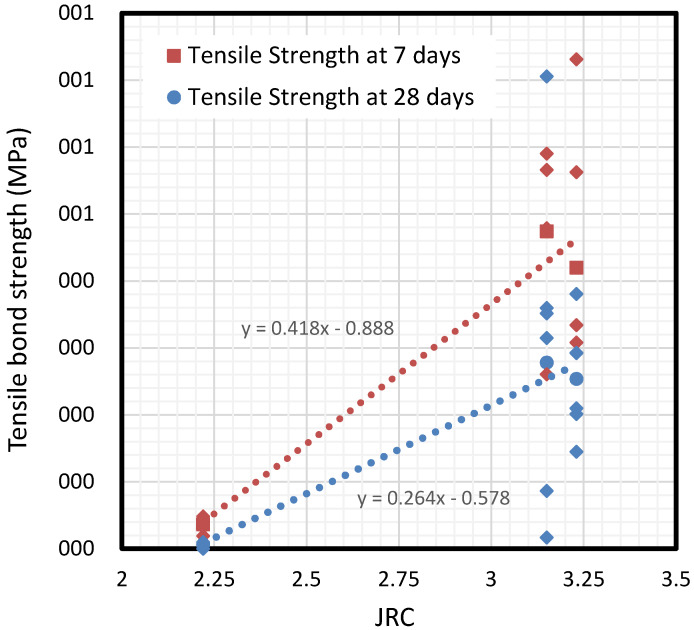
Tensile bond strengths depending on the roughness factor JRC.

**Figure 11 materials-13-02183-f011:**
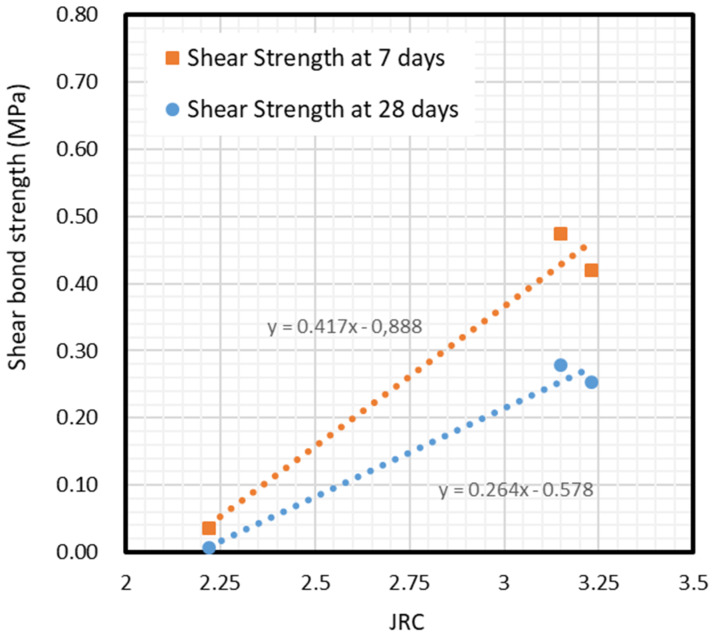
Comparison between cohesion at different bond strengths and JRC.

**Table 1 materials-13-02183-t001:** Properties of the four masonry materials studied in this project.

Material Code	fb ^1^ (N/mm^2^)	Absorption (%)	Density (kg/m^3^)
MP1	35	<6	1850
MP2	40	<6	2200
MP3	15	<12	1650
MP4	−	−	1900

^1^ Normalized compressive strength of masonry.

**Table 2 materials-13-02183-t002:** Sprayed mortar mixture.

Component	Details	Content (Ratio to Cement Mass)
Cement	CEM I 52.5R	1.00
Water	Tap water	0.51
Sand	Limestone, 0–1 mm	1.70
Plasticizer	Sika, Viscocrete 5920, modified polycarboxylate in aqueous base, 34% solids	0.01
Accelerator	IQE, ALNA73, sodium aluminate rich in alkalis	0.06

**Table 3 materials-13-02183-t003:** Mechanical properties of sprayed mortar.

Age	fc ^1^ (N/mm^2^)	Density (kg/m^3^)	Porosity (%)	Absorption (%)
7	21.97	1863.55	24.2	12.96
28	25.11	1866.68	24.18	12.93

^1^ Compressive strength of mortar.

**Table 4 materials-13-02183-t004:** Summary of specimen details and tests results.

Test Type	Sample Type	Effective Sample Number	JRC	Ra (mm)	JCS	Age (days)	Average Value (MPa)	CV
Tensile	MP1	4	3.23	0.42	26.99	7	0.48	0.41
	MP2	4	2.22	0.26	28.36	7	0.04	0.36
	MP3	4	3.15	0.31	17.83	7	0.47	0.32
	MP1	5	3.23	0.42	29.24	28	0.25	0.33
	MP2	3	2.22	0.26	30.85	28	0.01	0.45
	MP3 ^1^	3	3.15	0.31	18.78	28	0.28	0.46
Shear	MP2	4	2.22	0.26	17.83	7	0.09	0.23
	MP1 ^2^	4	3.23	0.42	29.24	28	0.43	0.23
	MP2	6	2.22	0.26	30.85	28	0.01	0.25
	MP3 ^3^	5	3.15	0.31	18.78	28	0.74	0.33

Legend: CV: coefficient of variation. ^1^ Five samples were successfully tested but only three gave valid results for the analysis. Two results were eliminated: one that was excessively large (MP3-C-C-8) and one that was excessively small (MP3-C-C-10). ^2^ Eight samples were successfully tested but only four gave valid results for the analysis. Four results were eliminated: two excessively large (MP1-A-B-6, MP1-A-B-7) and two excessively small (MP1-A-B-1, MP1-B-A-7). ^3^ Nine samples were successfully tested but only five gave valid results for the analysis. Four results were eliminated: two that were excessively large (MP3-C-C-9, MP3-C-B-2) and two that were excessively small (MP3-D-A-8, MP3-D-A-10).

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
