# Peer review of "Bond Strength Tests under Pure Shear and Tension between Masonry and Sprayed Mortar"

_materials, 2020, doi:10.3390/ma13092183_

Round 1

Reviewer 1 Report

The submitted manuscript presents an extensive experimental study to investigate the bond strength under pure shear and tension between masonry and sprayed mortar, without normal force or constraint. Three masonry materials were used.

The manuscript describes the experimental campaign, namely the materials properties and the samples preparation, as well as the performed experimental tests (roughness and pull-out testes). The results are presented and discussed, namely in light of the relations between the bond strength, mortar age, roughness and materials strength. In the end, a novel logarithmic relationship between the strengths was proposed for the type of used materials.

The topic that is developed in the study has novelty and is valuable, since it deals with a feasible and economical technique to reinforce structural masonry walls in existing buildings, consisting of simply spraying cementitious materials. In addition, a lack still exists in the literature on the study of bonding between masonry materials and sprayed mortar. The presented results and proposed relations can be useful for designers to analyse sprayed masonry walls.

I made some comments in order to improve the manuscript. The authors should take the comments into account and revise their manuscript.

Comment 1

The reading of the article must be reviewed, minor spell check is required. Some examples:

- line 29: correct “… particularly true of cities …” to “… particularly true for cities …”;

- line 132: correct “This a parameter …” to “This parameter …”;

- …

Comment 2

Some typos must be corrected throughout the manuscript. Some examples:

- line 97: symbol “fi” is not the same as in Eq. (1);

- footnote of Table 3: it should be “mortar” instead of “concrete”;

- Line 236: correct “Figure 5” to “Figure 4”

- Line 249: it should be “… shows the average curves values…”, because no curves are presented in Table A2.

- Section “Analysis and discussion”, and subsequent subsections, must be renumbered to “5” instead of “4”. Same for “Conclusions”.

- …

Comment 3

Line 189: What is “effect of diffraction on the test results”? I think it should be “spreading” instead of “diffraction” (?). Please clarify in the text or correct.

Comment 4

Fig. 3(b) should be removed from Fig. 3 and inserted into Fig. 4.

Comment 5

The symbols in the graphs of Fig. 3(c)-(d) are hard to read. Please enlarge such graphs.

Comment 6

Line 225: An “error!” statement appear in the pdf text for the reference. Please correct.

Comment 7

Line 235: Please clarify “as described in the following sections”. No additional sections exist in Section 3.2.1 to describe the direct shear an tensile tests. Please delete or correct the sentence.

Comment 8

In the graph of Fig. 6, the legend of the vertical axis should be named as “Tensile/Shear bond strength”

Comment 9

In all presented graphs throughout the article, a bar error must be inserted next to each presented average point. Average values have no meaning without the indication of the error range.

Comment 10

Line 314-315: It is stated that the “normalized compressive strength of masonry is not really significant to the bond behavior if the strengths of both components do not vary widely”. It is true for the tensile bond strength, however for the shear bond strength the difference is about 70% (from the graph in Fig. 6)! Please clarify or revise the sentence.

Comment 11

Fig. 8 and 9 should also present an additional graph for the shear bond strength, for comparison and discussion in the text.

Comment 12

For Eq. (7), (8) and (11) please indicate the value for R^2.

Comment 13

Based on your experimental results and used materials, the predictions from the proposed Eq. (9)-(10) and Eq, (1) should be compared and discussed.

Comment 14

In Tables A3 and A4, please explain, in a foot note, the meaning of the name of specimens and of failure mode.

Comment 15

References must be uniformized in the reference list. For instance, the presented name of authors in references [14] and [27] only includes initials.

Author Response

Dear Reviewer,

Thank you for giving us constructive suggestions which helped us to improve the quality of the paper. It has been reworked following the recommendations and suggestions made.

Please find attached the new version of the manuscript entitled "Bond strength tests under pure shear and tension between masonry and sprayed mortar" as well as the answers to each comment and indications of the modifications carried out on the manuscript. All changes in the main text are highlighted in orange. Some new citations have been included as suggested by the reviewers. We believe the changes made in the revised version have contributed to the improvement of the paper and are expected to fulfill the reviewers’ suggestions.

Yours sincerely,

Dawei Huang, Oriol Pons and Albert Albareda

-Reviewer 1

Comment 1

The reading of the article must be reviewed, minor spell check is required. Some examples:

- line 29: correct “… particularly true of cities …” to “… particularly true for cities …”; OP - Corrected

- line 132: correct “This a parameter …” to “This parameter …”; OP – Corrected

- …

  • We have fixed these mistakes and carried out a complete English Language and Spelling revision.

Comment 2

Some typos must be corrected throughout the manuscript. Some examples:

- line 97: symbol “fi” is not the same as in Eq. (1); OP - Corrected

- footnote of Table 3: it should be “mortar” instead of “concrete”; OP - Corrected

- Line 236: correct “Figure 5” to “Figure 4” OP - Corrected

- Line 249: it should be “… shows the average curves values…”, because no curves are presented in Table A2. OP - Corrected

- Section “Analysis and discussion”, and subsequent subsections, must be renumbered to “5” instead of “4”. Same for “Conclusions”. OP - Corrected

- …

  • We have fixed these corrections and revised again the manuscript for more mistakes.

Comment 3

Line 189: What is “effect of diffraction on the test results”? I think it should be “spreading” instead of “diffraction” (?). Please clarify in the text or correct.

  • We have added the following, so this part of the text is more understandable:

In order to reduce the effect of shrinkage of the entire body and avoid the defective zone from the spraying process [31] on the test results, no specimen was placed within the range of 10 cm of the surrounding edge, and the side face of the trough was made at 45 degrees against the base.

Comment 4

Fig. 3(b) should be removed from Fig. 3 and inserted into Fig. 4.

  • We have done this change and updated the citations to these figures.

Comment 5

The symbols in the graphs of Fig. 3(c)-(d) are hard to read. Please enlarge such graphs.

  • We have redrawn these graphs.

Comment 6

Line 225: An “error!” statement appear in the pdf text for the reference. Please correct.

  • This mistake appeared when generating the pdf file. We have fixed it in this new version of the manuscript.

Comment 7

Line 235: Please clarify “as described in the following sections”. No additional sections exist in Section 3.2.1 to describe the direct shear an tensile tests. Please delete or correct the sentence.

  • We have deleted this sentence.

Comment 8

In the graph of Fig. 6, the legend of the vertical axis should be named as “Tensile/Shear bond strength”

  • We have corrected it.

Comment 9

In all presented graphs throughout the article, a bar error must be inserted next to each presented average point. Average values have no meaning without the indication of the error range.

  • We have introduced it.

Comment 10

Line 314-315: It is stated that the “normalized compressive strength of masonry is not really significant to the bond behavior if the strengths of both components do not vary widely”. It is true for the tensile bond strength, however for the shear bond strength the difference is about 70% (from the graph in Fig. 6)! Please clarify or revise the sentence.

  • We have changed the sentence to make clear that this is only in tensile bond strength.

Comment 11

Fig. 8 and 9 should also present an additional graph for the shear bond strength, for comparison and discussion in the text.

  • We have added and discussed this additional graph.

Comment 12

For Eq. (7), (8) and (11) please indicate the value for R^2.

  • The values for R^2 have been specified.

Comment 13

Based on your experimental results and used materials, the predictions from the proposed Eq. (9)-(10) and Eq, (1) should be compared and discussed.

  • We share the interest of the reviewer on validating the results predicted by Eq. (9)-(10) and (1) but the experimental campaign which has been carried out does not include normal stress in any case (σ), so the experiments provide only values for cohesion (initial shear stress), independently of the conditions of normal pressure. This fact, that this experimental campaign does not include normal stress, is explained in the last paragraph of the introduction.

  • Comment 14

In Tables A3 and A4, please explain, in a foot note, the meaning of the name of specimens and of failure mode.

  • We have added it.

Comment 15

References must be uniformized in the reference list. For instance, the presented name of authors in references [14] and [27] only includes initials.

  • We understand the reviewer comment. Nevertheless, we have used the journal template and it automatically puts initials instead of names in these two cases. We think it is because there are more than three authors, and this are this reference style rules. In the previous version of the manuscript reference 27 had a name because there was a mistake in the BiBtex file. In any case, we have no problem in editing it and writing the complete names if it is all right with the Editor.

Reviewer 2 Report

This paper investigated the bond strength under pure shear and tension. The outcome is interesting to readers. However, there are several aspects that need to be improved. The reviewer can only recommend for publication if the author satisfactorily address the following comments in the revised version.

  1. Suggest to add scale in Figure 5. It is not clear how much area of the specimen is covered in the photo.
  2. What is the mechanism behind the higher shear bond strength than tensile bond strength in Figure 6?
  3. Figure 7 does not indicate which one is the specimen MP1, MP2 and MP3? Why there is a significant drop of JCS at 40 MPa?
  4. Key findings from section-4 need to be highlighted in conclusion section.
  5. The introduction section has not provided sufficient information on the relationship between shear and tensile strength. Recent investigation has shown that tensile strength is correlated with shear resistance of the concrete [Ref: Optimal design for epoxy polymer concrete based on mechanical properties and durability aspects + Effect of elevated in-service temperature on the mechanical properties and microstructure of particulate-filled epoxy polymers].

Author Response

Dear Reviewer,

We would like to thank you for giving us constructive suggestions which helped us to improve the quality of the paper. It has been reworked following the recommendations and suggestions made.

Please find attached the new version of the manuscript entitled "Bond strength tests under pure shear and tension between masonry and sprayed mortar" as well as the answers to each comment and indications of the modifications carried out on the manuscript. All changes in the main text are highlighted in orange. Some new citations have been included as suggested by the reviewers. We believe the changes made in the revised version have contributed to the improvement of the paper and are expected to fulfill the reviewers’ suggestions.

Yours sincerely,

Dawei Huang, Oriol Pons and Albert Albareda

-Reviewer 2

This paper investigated the bond strength under pure shear and tension. The outcome is interesting to readers. However, there are several aspects that need to be improved. The reviewer can only recommend for publication if the author satisfactorily address the following comments in the revised version.

1- Suggest to add scale in Figure 5. It is not clear how much area of the specimen is covered in the photo.

  • We understand the reviewer comment and to solve this problem we have added a scale in the pictures.

2- What is the mechanism behind the higher shear bond strength than tensile bond strength in Figure 6?

  • The mechanism is through the roughness of surfaces (mechanical interlock), which leads to a brittle fracture for low-bearing masonry that ends in more shear capacity rather than the tensile one. From obtained results, we know that JRC is a crucial parameter in the bond response of sprayed mortar on masonry surfaces; but this is especially significant in case of shear due to the influence of mechanical interlock. The authors have explained so in lines from 325 to 329 of Section 5.1.

3- Figure 7 does not indicate which one is the specimen MP1, MP2 and MP3? Why there is a significant drop of JCS at 40 MPa?

  • This has been added to Figure 7. The reason why there is a drop of JCS at 40 MPa specimens is because there is a higher difference in strengths between the two materials. While sprayed mortar is about 25 MPa, the masonry base is 40 MPa strength. This fact, with a difference of 15 MPa, leads to a brittle fracture of the bond, and therefore, a lower JCS parameter. This has been further explained in lines from 321 to 325 of Section 5.1.

4- Key findings from section-4 need to be highlighted in conclusion section.

  • The authors have highlighted in the conclusions some findings from the discussion.

5- The introduction section has not provided sufficient information on the relationship between shear and tensile strength. Recent investigation has shown that tensile strength is correlated with shear resistance of the concrete [Ref: Optimal design for epoxy polymer concrete based on mechanical properties and durability aspects + Effect of elevated in-service temperature on the mechanical properties and microstructure of particulate-filled epoxy polymers].

  • The authors agree that the research paper can improve this issue. In this sense we have added the following sentences and references in the third paragraph:
  • "On the other hand, focusing on the mechanical properties of cementitious materials, recent studies have shown that tensile strength is correlated with shear resistance. A study showed that the shear strength is generally greater than tensile strength in the specific case of epoxy polymer concrete [21]. Another study proved that the shear strength is almost equal the tensile strength in the case of low normal stress [16]."

Round 2

Reviewer 1 Report

I received and read the revised version of the article “Bond strength tests under pure shear and tension between masonry and sprayed mortar”. I´m generally satisfied with the authors´ replies to my earlier comments and I also consider that most of my suggestions and concerns have been properly explained and considered by the authors to improve the article.

I consider that the revised article submitted by the authors can be accepted in the present form to be published.